# Strategic and Technical Considerations in Manufacturing Viral Vector Vaccines for the Biomedical Advanced Research and Development Authority Threats

**DOI:** 10.3390/vaccines13010073

**Published:** 2025-01-14

**Authors:** Lindsay A. Parish, Shyam Rele, Kimberly A. Hofmeyer, Brooke B. Luck, Daniel N. Wolfe

**Affiliations:** 1CBRN Vaccines, Biomedical Advanced Research & Development Authority (BARDA), Administration for Strategic Preparedness and Response (ASPR), U.S. Department of Health and Human Services (HHS), Washington, DC 20201, USA; 2Division of Research, Innovation, and Ventures (DRIVe), Biomedical Advanced Research & Development Authority (BARDA), Administration for Strategic Preparedness and Response (ASPR), U.S. Department of Health and Human Services (HHS), Washington, DC 20201, USA; 3Pharmaceutical Countermeasures Infrastructure (PCI) Division, Biomedical Advanced Research & Development Authority (BARDA), Administration for Strategic Preparedness and Response (ASPR), U.S. Department of Health and Human Services (HHS), Washington, DC 20201, USA; 4CBRN, Biomedical Advanced Research & Development Authority (BARDA), Administration for Strategic Preparedness and Response (ASPR), U.S. Department of Health and Human Services (HHS), Washington, DC 20201, USA

**Keywords:** viral vectors, vaccine manufacturing, vaccine manufacturing sustainability, filoviruses, BARDA, Biomedical Advanced Research and Development Authority

## Abstract

Over the past few decades, the world has seen a considerable uptick in the number of new and emerging infectious disease outbreaks. The development of new vaccines, vaccine technologies, and platforms are critical to enhance our preparedness for biological threats and prevent future pandemics. Viral vectors can be an important tool in the repertoire of technologies available to develop effective vaccines against new and emerging infectious diseases. In many instances, vaccines may be needed in a reactive scenario, requiring technologies than can elicit rapid and robust immune responses with a single dose. Here, we discuss how viral vector vaccines are utilized in a vaccine portfolio for priority biological threats, some of the challenges in manufacturing viral vector vaccines, the need to strengthen live virus manufacturing capabilities, and future opportunities to capitalize on the use of viral vectors to improve the sustainability of the Biomedical Advanced Research and Development Authority’s vaccine portfolio.

## 1. Introduction

The mission of the Biomedical Advanced Research and Development Authority (BARDA)—part of the Administration for Strategic Preparedness and Response (ASPR) within the U.S. Department of Health and Human Services (HHS)—is to develop medical countermeasures that address the public health and medical consequences of chemical, biological, radiological, and nuclear (CBRN) accidents, incidents, and attacks, pandemic influenza, and emerging infectious diseases. As part of this mission, BARDA has cultivated a portfolio of investments in vaccine development for biological threats such as anthrax, smallpox, and filoviruses [1].

In general, there are six different types of vaccines that each utilize different strategies to elicit an immune response to a whole or part of a pathogen including live attenuated, inactivated, subunit, toxoid, nucleic acid, and viral vector vaccines [2]. Each vaccine type has advantages and disadvantages, and the selection of one for development should take into consideration factors such as immunogenicity, number of doses, durability, reactogenicity, cost of goods, and the ease and scale of manufacturing. Live attenuated vaccines have been the foundation of vaccinology from the earliest smallpox efforts. While often effective, these approaches can involve substantial risk when the virus in question causes high levels of morbidity and mortality. For example, attenuation of an Ebola strain for use as a vaccine would require initial production under BSL-4 constraints and comprehensive data packages to ensure an acceptable toxicity profile prior to moving into clinical development. For these reasons, viral vectors have been utilized in vaccine development and represent key investments for such diseases that require high level containment for wildtype strains.

Some additional key general advantages of viral vector vaccines include the ability to elicit both humoral and cellular immunity, production of a robust immune response, and the flexibility of viral vectors in general to encode a wide range of antigens from different pathogens (reviewed in [3]). As live attenuated and viral vector vaccines most closely mimic a natural viral infection by infecting host cells, these types of vaccines are often able to elicit a robust and durable response with a single vaccine dose. A rapid onset of robust protective immunity with a single dose is essential for vaccines that will be used in a reactive outbreak setting against high-consequence pathogens. This is reflected in current BARDA programs to develop vaccines for Ebola (EBOV), Sudan virus (SUDV), and Marburg virus (MARV), which have all utilized viral vector vaccine technologies [1]. 

Viral vector vaccines usually consist of an attenuated or nonpathogenic virus, which encodes a heterologous antigen from a pathogen of interest. This heterologous antigen can be presented in the vector vaccine virion or encoded in the viral DNA or RNA and then expressed in the host cell. Moreover, viral vector vaccines can be subdivided into replicating and nonreplicating vectors. While nonreplicating vectors are perceived to be safer, they may require a higher titer or multiple doses to achieve the desired immune response (reviewed in [3,4]). Likewise, replicating vectors with the ability to multiply themselves and the antigen of interest inside the host cell can often generate robust immune responses in a single dose as was seen in the Ebola virus vaccine ERVEBO [5]. However, as with other live vaccines, viral vector vaccines have the potential for unique reactogenicity challenges and often have complex requisite manufacturing processes. Target product profiles (TPPs) for BARDA viral vector vaccines, including risk-benefit profiles and the discussion of safety and reactogenicity have been reviewed in [6], and BARDA threat-specific TPPs can be found online [7].

Here, we highlight key considerations that inform a strategic approach for how BARDA could improve the sustainability of a growing portfolio of viral vector vaccine technologies despite the complex manufacturing requirements. Considerations include BARDA’s investments in viral vector vaccines, general challenges associated with manufacturing viral vectors, ways to mitigate some of these challenges, and the national security need to maintain and increase domestic live viral vaccine manufacturing.

## 2. BARDA’s Portfolio of Viral Vector Vaccines

### 2.1. Filovirus Viral Vector Vaccines

To combat filovirus threats, BARDA has supported the advanced development of multiple vaccine candidates against EBOV, SUDV, and MARV, all using viral vectors (Table 1) (reviewed in [8]).

A prime example of the utility of viral vector vaccines for high-consequence pathogens is ERVEBO, which was licensed by the Food and Drug Administration (FDA) in 2019 for the prevention of Ebola virus disease. In 2023, the indication for ERVEBO was expanded to include children one year of age and older. This vaccine is based on the vesicular stomatitis virus (VSV) to express the EBOV glycoprotein as a live replicating vector vaccine. ERVEBO was demonstrated to be an effective vaccine as a single dose during a randomized ring vaccination study during the 2014–2016 West Africa Ebola outbreak [17,18]. ERVEBO has since been deployed to multiple smaller EBOV outbreaks in sub-Saharan Africa and aided in preventing these smaller outbreaks from reaching the magnitude of the 2014–2016 West Africa Ebola outbreak. Additionally, ERVEBO was determined to elicit durable immune responses, as antibodies were still detectable in adults and children one year after vaccination [19]. Given the success of Merck’s ERVEBO as an EBOV vaccine, the same underlying technology and VSV vector is being applied to MARV and SUDV vaccine development by multiple companies [20].

Other vectors that have been used in the development of EBOV vaccines include a heterologous EBOV vaccine utilizing two different viral vectors, Adenovirus 26 (Ad26) and Modified Vaccinia virus Ankara (MVA). This vaccine, which was licensed as Zabdeno and Mvabea by the European Medicines Agency (EMA) in 2020, involves a two-dose regimen of Ad26.ZEBOV followed by MVA-BN-Filo approximately eight weeks later. While Zabdeno/Mvabea vaccine does require two doses for EBOV, the MVA vector also encodes the SUDV and MARV glycoproteins as well as the Tai Forest virus nucleoprotein. Currently, the Zabdeno/Mvabea vaccine is indicated only for EBOV [21,22]. Chimpanzee Adenovirus 3 (ChAd3)-vectored vaccines are in development as two monovalent vaccines against MARV and SUDV, an approach that had previously been applied to EBOV and advanced into a Phase 2 study during the West Africa epidemic [23]. As of this writing, the ChAd3-MARV and ChAd3-SUDV vaccines are in Phase 2 clinical studies in Uganda and Kenya [24,25]. Recently, the ChAd3-MARV vaccine was shown to demonstrate 100% protection in cynomolgus macaques [26].

### 2.2. Project NextGen and COVID-19 Viral Vector Vaccines

In addition to filoviruses, the COVID-19 response efforts have involved multiple vaccines utilizing viral vectors. For instance, the Ad26-vectored COVID-19 vaccine, is based on the same Ad26 vector as their EBOV vaccine. While the Emergency Use Authorization (EUA) from the FDA for the COVID-19 vaccine was withdrawn in May 2023 [27], this vectored vaccine demonstrated a 66.3% effectiveness against laboratory-confirmed COVID-19 in a single dose [28]. Likewise, the COVID-19 vaccine based on the ChAdOx-1 vector was initially estimated at 74% effective at preventing symptomatic disease 15 days or later after the second dose [29]. More than 2 billion doses of the ChAdOx-1 COVID-19 vaccine were delivered globally [30], although it has since voluntarily withdrawn this vaccine from the market [31].

Project NextGen, a collaboration between BARDA and the National Institute of Allergy and Infectious Disease (NIAID), is advancing the clinical testing of next-generation vaccine candidates that have the potential to offer improved durability, breadth, and transmission blocking compared to the currently available COVID-19 vaccines [32]. Mucosal administration is an important strategy to achieve these goals, as generation of immune responses at the site of infection has the potential to improve vaccine effectiveness, particularly infection and transmission blocking. Viral vectors are a promising vaccine technology for mucosal administration given their natural ability to cross the harsh mucosal barrier and generate robust mucosal responses, which is currently a challenge for nucleic acid-based approaches [33].

Under Project NextGen, one goal is to support Phase 2b clinical trials for several next generation mucosal COVID-19 vaccines that use viral vector technologies. This includes an intranasal NDV-HXP-S candidate based on the Newcastle disease virus (NDV) vector; an intranasal CVXGA candidate based on the parainfluenza virus 5 (PIV5) vector; and an oral pill candidate based on the Adenovirus type 5 (Ad5) vector—all of which express the SARS-CoV-2 spike protein [34]. An intramuscular candidate, GEO-CM04S1, that is based on the MVA vector and encodes for both spike and nucleoprotein of SARS-CoV-2 in a single vector is also in development [35]. Similarly, the MVA vector-based GEO-CM04S1 candidate was originally developed as a COVID-19 vaccine option for use in immunocompromised people and is currently in two Phase 2 trials for chronic lymphocytic leukemia patients and for stem cell transplant patients [36,37]. 

## 3. Challenges with Viral Vector Vaccine Manufacturing

While viral vector vaccines may offer advantages of eliciting a robust and durable immune response and potential utility for mucosal approaches, the manufacturing process can be complex with distinct requirements varying between different viral vectors. One major challenge that contributes to increased costs of viral vector vaccine manufacturing is that live viral vectors require the use of manufacturing facilities that have Biosafety Level 2 (BSL-2) capability, which entails using facilities designed for containment including unidirectional flow of materials and waste, as well as additional safety measures to protect staff [38,39]. For instance, a NDV-vectored COVID-19 vaccine candidate requires BSL-2 conditions to grow the virus to high concentrations [40]. Moreover, some contract manufacturing organizations (CMOs) may opt not to have multiple products manufactured in a single facility and instead opt to have dedicated facilities or production suites for manufacturing vaccines using a particular vector to avoid cross contamination of products [41]. However, dedicated facilities and equipment preclude other products from being manufactured and thus add to the overall costs of viral vector vaccine production. 

Similarly, other challenges that can lead to increased costs are the varying upstream and downstream processes involved in manufacturing, which can take significant time to optimize. Viral vectors are usually matched with a particular mammalian cell line for optimized viral growth, which may not be interchangeable between viral vectors. A variety of different cell types can be used for vaccine manufacturing including but not limited to Vero cells, Procell 92.S, HEK, and CHO cells that can be adapted to large-volume cultures [8,42]. Furthermore, downstream processes such as filtration and purification steps can be different from vector to vector and may require extensive process optimization to achieve the desired yields. These processes include purification of the vaccine virus from host cell material and empty capsids lacking viral genomic material, by utilizing methods such as affinity or ion-exchange chromatography [43,44]. 

### 3.1. Analytical Testing and Product Quality

Critical to the success of the GMP manufacturing of viral vector vaccines is the ability to measure, quantify, and validate the potency of the vectored vaccines as a key quality metric to demonstrate to batch-to-batch product consistency, product lot release, and vaccine stability at the desired scales. Potency and supporting orthogonal assays that can quantitatively measure (a) the expression of the transgene (payload) to deliver the functionally active target antigen and (b) the vector genome (vg) or viral particle (vp)/mL and address product variability and differences in clinical efficacy (lot-to-lot, plus change in manufacturing processes, sites, or contract manufacturers) remains a pain point for regulatory assessment and approval. Optimizing potency assays and vector characterization not only impacts the ability to develop the viral vector vaccine but is also directly linked to the safe dose range determination purposes, as viral particle (vp)/mL or viral titers (total and infectious) for both replication-competent and replication-deficient viral vectors are universally used for dosing decisions in both preclinical and clinical studies [45].

Integration of high-throughput methods of measuring the viral titer and developing real-time analytics at appropriate steps to detect product quality (including host cell impurities) by in-process testing and during downstream purification steps can serve as crossover learnings for various product characterization and release assays of viral vaccine drug substances. Moreover, there is a current lack of standardization in multiple processes involved in viral vector manufacturing. The resulting risks such as the variability between products, titering methods, and viral-genome load versus capsid exposure can be mitigated by the use of reference standards that can reduce the variability giving more precise and reliable quantification. Additionally, it would be ideal if the production of various viral vector vaccines could be adapted to suspension cell culture and process intensification/perfusion culture methods. In these systems, cell culture is optimized at high cell densities upstream to produce higher viral harvest titers using the same process equipment and input/output modules, offering a strategy to produce larger numbers of doses in smaller production facilities worldwide. There will be a trade-off between yield and impurities that will have to be evaluated on a case-by-case basis for different vectors. Adaptation of cell culturing processes to suspension cell culture and process intensification/perfusion culture methods for different vectors can aid in expediting manufacturing development timelines and enable rapid production of doses within smaller (such as pilot-scale size) facilities. Overall, optimization of analytical testing and cell culture methods for one viral vector vaccine has the potential to reduce the development and manufacturing risks of other vaccines utilizing the same vector.

Finally, like nucleic acid vaccines, after fill-finishing, some viral vector vaccines require ultra-cold temperatures for storage and transport. Storage and shipping of doses at these temperatures increases the costs of maintaining an inventory of vaccine doses as well as the operational use of vaccines. Additional investments in thermostable formulations to enhance the stability of viral vector vaccines at the desired storage and transport temperatures may aid in reducing the costs of maintaining and shipping inventory of some viral vector vaccines.

### 3.2. Strategies to Optimize and Maintain a Portfolio of Viral Vector Vaccines

To optimize a portfolio of investments in viral vector vaccines, experience in manufacturing one viral vector vaccine has the potential to carry over and reduce the time and costs in the development and manufacturing of other vaccines using the same viral vector. Leveraging the same vector for multiple vaccine candidates enables a continuous active approach to “warm basing”, whereby a steady cadence of manufacturing to meet demands for different vaccines collectively supports the sustainability of the capability. A dedicated manufacturing suite for candidates using the same vector could, therefore, reduce the facility and equipment investment costs. Additionally, such a “continuous active” warm basing approach could potentially allow for investments in manufacturing capabilities that can be utilized for surge capacity and rapid scale-up during an outbreak [46]. This scenario has potential as there are now multiple vaccine candidates for a variety of threats using the VSV or adenovirus vectors [47,48]. Similarly, experience with a particular cell line and viral vector may also allow for utilizing the same or similar upstream and downstream manufacturing processes and analytical testing toolbox thus reducing the time and development costs. 

While the mRNA platform is considered an end-to-end platform for vaccine production, the viral vectors can be considered as semi-platforms that require customization of certain steps based on the individual viral vector vaccine target, especially when it comes to the downstream purification. However, there are some common elements that could be leveraged for further acceleration opportunities and shorter development timelines. The use of consolidated platforms and prior historical knowhow for the rapid execution of developability assessments, formulation studies, and process development can be valuable in this regard, for example, (i) genericizing the upstream process and certain purification unit operations (e.g., the method for clarification, viral inactivation, and filtration); (ii) predefining high throughput resin screening in parallel and combining scale-down chromatography experimentation to accelerate chromatography resin selection; (iii) defining critical process parameters and technical risk assessments based on prior historical knowledge; (iv) platforming approaches to small-scale model qualification, process characterization, and technology transfer between development, GMP, and commercial manufacturing teams; (v) implementation of phase-appropriate viral clearance strategy and standardizing testing of adventitious agents for different vectors; (vi) leveraging platform and pre-verified/QA-approved raw materials for viral vector development and scaling-up; and (vii) accelerated stability modeling to support supply chain management with respect to the monitoring of the quality of vaccines during shipping, especially by estimating the impact of potential cold-chain breaks.

Strategies that leverage this approach to sustaining the manufacturing capacity for a particular viral vector should assess the potential risk and impact of anti-vector immunity. Vaccination with one viral vector candidate could result in an immune response not only to the pathogen of interest but also to the viral vector backbone that could interfere with eliciting an immune response to the second pathogen of interest. This risk varies across viral vector types; for example, pre-existing adenovector immunity reduced antigen-specific immunity for Ad5-vectored HIV vaccines, whereas in preclinical studies pre-existing MVA immunity did not impact the overall efficacy of an MVA-based vaccine [49,50]. Pre-existing immunity may also be less of a concern in vectors where the approach is to replace the native immunodominant surface glycoprotein of the vector virus with that of the target virus. This risk may be further mitigated, as different vaccine candidates based on the same platform may not necessarily have the same target population, e.g., currently EBOV and Nipah virus are geographically distinct. 

## 4. Limited Available U.S. CMOs with Capability for Live Virus or Viral Vector Manufacturing and Fill Finish

The COVID-19 pandemic highlighted the importance of having a national capability to quickly manufacture and fill vaccine doses. During the pandemic, over 300 COVID-19 vaccine candidates were in development worldwide [51]. The vaccines which were eventually licensed or granted EUAs generally were successful in manufacturing vaccine doses quickly [52]. Thus, having the capability and access to manufacture doses was one of the most important factors in moving vaccine doses out quickly to a population. It is difficult to predict which type of vaccine and technology would be successful against a new pathogen; therefore, maintaining manufacturing capability for multiple vaccine technologies is essential to our national security in order to be prepared to respond to the next pandemic. Live virus vaccines are one, albeit important, category of vaccine technologies that require specific manufacturing facilities as well as experienced staff. There are currently a limited number of US CMOs that have the end-to-end capacity for live virus or viral vector manufacturing in a BSL-2 environment, including bulk drug substance manufacturing as well as fill-finish capabilities for drug product, at the appropriate scale. Additionally, phase appropriate manufacturing of viral vaccines will require different partners at different phases of clinical development and commercial production. Partners suitable for Phase 1 activities may not have the appropriate capacities and capabilities for later stage development and commercialization purposes. Bottlenecks exist in viral vector production not only for infectious disease vaccines, which have a low to no commercial market, but also for cancer vaccines and gene therapy products, as these product types compete with each other for access to CMOs for manufacturing [53,54]. This includes plasmid production in the US, which is also a bottleneck vis-à-vis an already committed clientele advancing other biologics and therapeutic modalities including monoclonal Abs, mRNA vaccines, and a high demand for cell and gene therapy products. Collectively, these bottlenecks limit capacity in the US with long wait times, which can derail domestic pandemic response capability. However, this competition also means that there is potential to strategically leverage multiple healthcare spaces to strengthen and sustain domestic manufacturing capacity for viral vector technologies.

## 5. Biopharmaceutical Manufacturing Preparedness (BioMaP)

As was seen during the COVID-19 pandemic, future public health emergencies may require medical countermeasures to be manufactured rapidly and at a scale to cover the entire U.S. population. Based on lessons learned from the COVID-19 response, in order to decrease time for vaccine production for the U.S. population and globally, the US has established the ambitious goal of the “100 day mission” [55]. To meet this goal, commercial-scale manufacturing will need to be sustained and available for the next pathogen that poses a potential global pandemic. In order to strengthen the domestic manufacturing capabilities of medical countermeasures, including live viral and viral vector vaccines, BARDA created the Biopharmaceutical Manufacturing Preparedness (BioMaP) Program. The BioMaP program consists of three main pillars essential to biomanufacturing pandemic preparedness and response including the BioMaP-Exercise (BioMaP-X), which identifies constraints and mitigates risks in the biomanufacturing process through routine exercises of capabilities, BioMaP-Workforce (BioMaP-W) which engages industry, academia, and trade groups to assess biomanufacturing workforce training capabilities, and the BioMaP-Consortium. The BioMaP-Consortium, composed of industry and academia partners from across the supply chain from manufacturers of critical raw materials and consumables to fill-finish services, has the goal of building a resilient, flexible, and surge-capable manufacturing supply chain [56]. As of this writing, the consortium currently has over 259 members including 11 nonprofit members and five institutions of higher education. Out of the 259 consortium members, only six had BSL-2 live viral manufacturing and or fill-finish capabilities (internal market research). In addition to convening industrial partners, the BioMaP-Consortium is intended to be a partnership vehicle from which consortium members will be able to respond to request for project proposals (RPPs) across BioMaP-Consortium’s three key domain areas: (1) industrial base expansion of the biomanufacturing supply chain; (2) biomanufacturing capacity expansion and reservation, and (3) advanced biomanufacturing technologies [57]. Recently, to inform a strategic approach to improving preparedness for domestic production of live viral vaccines, BioMaP-X held a biomanufacturing capability assessment (BCA) to determine the steps and potential gaps for a technology transfer package to domestically fill finish a vaccine whose drug substance was produced internationally. The major take-away from this BCA was confirmation that tech transfer of a live virus vaccine drug substance manufactured ex-U.S. to a U.S.-based CMO is a significant undertaking that requires the consideration of multiple factors including shipping, supply chain, contracts, regulatory, and quality. Preparing for these challenges will accelerate the delivery of medical counter measures when needed. Continued investments in domestic manufacturing capabilities and the biomanufacturing supply chain, particularly for live viral vaccines, will be needed for optimal preparedness for future biological threats. Subsequent biomanufacturing capability assessments will be conducted by BioMaP-X to continually evaluate the domestic biomanufacturing preparedness.

## 6. Discussion and Future Directions

Vaccines are a fundamental part of an overall preparedness and response strategy for infectious disease threats. During the COVID-19 pandemic, Phase I clinical trials were started approximately 2 months after the SARS-CoV-2 sequence became publicly available [58]. mRNA vaccines were successfully produced at commercial scale approximately 10 months from the declaration of a Public Health Emergency but only with significant USG investments or guaranteed advanced purchase agreements. Given the speed at which mRNA vaccines can be manufactured, it may be possible to maintain a smaller number of doses for a potential emergency and manufacture additional doses as needed. However, it has yet to be seen whether mRNA vaccines are capable of eliciting a robust immune response in a single dose as well as provide a fast onset of protection to filoviruses or other high-consequence pathogens in clinical studies. Additionally, as the demand for COVID-19 vaccines has waned in recent years, the manufacturing capabilities to produce mRNA vaccines have decreased to meet market demand.

While this discussion specifically focuses on viral vector vaccines, it is important to note that new technologies such as self-amplifying RNA (saRNA) vaccines show promise in both the speed of manufacturing [59] and potential for single-dose protection in small animal preclinical studies against the highly lethal EBOV and MARV [60]. Elsewhere, a COVID-19 saRNA booster vaccine received emergency use authorization in India in 2022, a second COVID-19 saRNA booster vaccine (ARC-154) received full approval in Japan in 2023, and several other saRNA vaccines for COVID-19 and influenza are currently in clinical development ([59,61], reviewed in [62]). Similar to live viral vaccines, saRNA vaccines, also as known as replicons, encode replication machinery that allows for the replication of the RNA and the encoded antigen of interest once inside a cell. Thus, saRNA vaccines generally require smaller doses and may be able to elicit a protective immune response in a single dose. Future studies on the effectiveness and onset of protection of mRNA and saRNA vaccines for filoviruses and other high-lethality pathogens should be investigated. 

Even with the promise of mRNA technologies, viral vectors address a critical niche in vaccine preparedness by providing a technology with a demonstrated potential for rapid single dose protection against infectious threats with severe morbidities and high mortality rates. With a growing number of licensed vaccines and multiple vaccine candidates in the development pipeline, BARDA is looking to devise strategies to sustainably maintain vaccines for multiple threats. Currently, there are multiple filovirus vaccine candidates coalescing around several viral vector families including VSV and adenovirus vectors. There are also multiple other high-consequence infectious diseases with vaccine candidates in development using VSV or adenovirus vectors, including Nipah virus disease, Lassa Fever, Middle East Respiratory Syndrome, Crimean Congo Hemorrhagic Fever, and Rift Valley Fever [63,64,65]. 

Planning for these vaccine candidates successfully reaching licensure should consider strategies that ensure viral vector manufacturing capacity is present and sustained via active use. Vaccine development using these technologies will ideally be informed by a thorough understanding of the existing domestic capacity for viral vector manufacturing, including both production of bulk drug substance product and fill/finish for the final product, as well as vector-specific technical considerations like the potential for single-dose protection and the impact of preexisting vector immunity. This along with the landscape of viral vector vaccines that are approved or in advanced development present several portfolio-based strategies to consider that take an active and holistic view on the sustainment of viral vector manufacturing capacity:Investments in higher-quality production of viral vector vaccines, reproducibility and scalability, maximizing facility output, and increasing overall process efficiency thereby enabling faster time from sequence to available doses and more doses per batch are important strategic drivers to improve preparedness and reduce the cost of goods;Collective sustainment of the manufacturing of a specific viral vector platform (or several different platforms if pre-existing vector immunity is a concern) across multiple different vaccine candidates; for example, sustaining VSV vector capacity collectively through filovirus vaccines and other high-consequence threats, like Lassa or Nipah viruses;Collective sustainment of the manufacturing for a specific manufacturing platform or cell line that is used across multiple types of viral vector or other vaccines; e.g., Vero cells are used for VSV vector vaccines and the FDA approved Chikungunya virus vaccine, Ixchiq [66];Collective sustainment of general BSL-2 manufacturing and fill/finish capacity across multiple viral vector vaccines for infectious diseases, with the potential to leverage other therapeutic areas (e.g., cancer vaccines and gene therapies);Reserving a fill line with a CMO with the experience and facilities necessary for live virus manufacturing.

In conclusion, viral vectors are being developed for a number of threats and in some cases used as licensed products, particularly for threats such as filovirus, where a robust immune response from a single dose is desired. Sustainment of a broad portfolio of licensed products is going to represent a major challenge where the commercial market is unpredictable at best and non-existent at worst. As global investments begin to coalesce around a few viral vectors, strategies such as warm-based manufacturing experience, equipment, and trained personnel, as well as reducing the challenges of costs associated with viral vector vaccine development and manufacturing become a feasible option for sustainment. These strategies will, of course, require investigation of the challenges of anti-vector immunity and licensing agreements with product developers. For example, vaccine developers working on threats where there is a very limited to no commercial marker for the indication may have licensed a vaccine technology from a larger corporate entity. This may impact or create constraints with respect to CDMO or vendor selection when implementing a strategy to sustain the viral vector manufacturing capacity and capability in this threat space. Overall, viral vectors are an important tool in the repertoire for vaccine development for a variety of threats. Increasing our U.S. capacity for live virus vaccine manufacturing and filling is essential for an optimal preparedness and response strategy for the next pandemic or public health emergency. 

## Figures and Tables

**Table 1 vaccines-13-00073-t001:** Summary of current BARDA viral vector vaccines.

Target	Viral Vector	Enveloped/Non-Enveloped	Genome	Payload Kilobases (kb)	Particle Size (nm)	Development Status
EBOV	VSV [9]	Enveloped	ss(-)RNA	~6–11 kb	bullet-shaped, 65 × 180 nm	Licensed; procurement
MARV	VSV					Phase 1 [10]
						Preclinical/
SUDV	VSV					Phase 1 [11]
MARV	ChAd3 [12,13]	Non-enveloped	dsDNA	~8–36 kb	85–100 nm	Phase 2
SUDV	ChAd3					Phase 2
COVID-19	NDV [14]	Enveloped	ss(-)RNA	~4 kb	~100 nm	Phase 2
COVID-19	PIV5	Enveloped	ss(-)RNA	~10 kb	100 to 200 nm	Phase 2b [15]
COVID-19	MVA [16]	Enveloped	dsDNA	~25–30 kb	~50 nm	Phase 2
Ad5	Ad5 [12,13]	Non-enveloped	dsDNA	8 kb (replication defective) 30 kb (helper dependent)	85–100 nm	Phase 2b

## Data Availability

No new data were created or analyzed in this study. Data sharing is not applicable to this article.

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
