# Peer review of "Strategic and Technical Considerations in Manufacturing Viral Vector Vaccines for the Biomedical Advanced Research and Development Authority Threats"

_vaccines, 2025, doi:10.3390/vaccines13010073_

Round 1
Reviewer 1 Report
Comments and Suggestions for Authors
This paper reviews how viral vector vaccines utilized in a vaccine portfolio for priority biological threats, some of the challenges in manufacturing viral vector vaccines, and discusses future opportunities of using viral vectors to improve the sustainability of the Biomedical Advanced Research and Development Authority’s vaccine portfolio. However, this manuscript is not recommended for journal publication for the following reasons:
1) Lack of technical novelty, scientific justification, and new insights beyond what is already known to viral vector-based vaccine area.
2) The authors claim to discuss challenges and opportunities in manufacturing viral vectors. However, very limited manufacturing technical points were discussed. The authors need to deeply discuss the manufacturing: upstream production, downstream purification, analytical method development, formulation, storage and shipping etc.
3) Also, need to clearly guide the audience, possibly consider classifying the viral vectors into enveloped viral vector, non-enveloped viral vector. What the bottlenecks, and current routes and future directions for each type of viral vectors manufacturing. Also, need to critically compare with non-viral vector vaccine, e.g. the LNPs
Overall, this manuscript is a good-quality technical report/guideline, but it does not meet the bar for a publishable journal article.
Author Response
Reviewer 1
This paper reviews how viral vector vaccines utilized in a vaccine portfolio for priority biological threats, some of the challenges in manufacturing viral vector vaccines, and discusses future opportunities of using viral vectors to improve the sustainability of the Biomedical Advanced Research and Development Authority’s vaccine portfolio. However, this manuscript is not recommended for journal publication for the following reasons:
- Lack of technical novelty, scientific justification, and new insights beyond what is already known to viral vector-based vaccine area.
- The authors claim to discuss challenges and opportunities in manufacturing viral vectors. However, very limited manufacturing technical points were discussed. The authors need to deeply discuss the manufacturing: upstream production, downstream purification, analytical method development, formulation, storage and shipping etc.
Response to points #1 and #2: Thank you for your comments and the note on the level of deep technical detail is well appreciated. The manuscript has been reframed in key parts to more clearly convey the intention of manuscript – which is to communicate the important role that viral vector vaccines play in a portfolio focused on very high-consequence threats (e.g., filoviruses) and key considerations that inform a strategic approach to ensure this vaccine portfolio – and viral vector vaccine capacity – is sustainable. The author’s goal is to relay to stakeholders how BARDA is thinking about this important preparedness question. Further, we have added additional technical details per the reviewer recommendations.
- Also, need to clearly guide the audience, possibly consider classifying the viral vectors into enveloped viral vector, non-enveloped viral vector. What the bottlenecks, and current routes and future directions for each type of viral vectors manufacturing. Also, need to critically compare with non-viral vector vaccine, e.g. the LNPs.
Response: Thank you for your comments. More text has been added to discuss bottlenecks for viral vector manufacturing. Additionally, Table 1 has been added to give more background on viral vectors/vaccines discussed and their current stage of clinical development. As this paper has been reframed to discuss the important role viral vectors play in a portfolio of vaccines against priority biological threats we have non subdivided bottlenecks between each type of viral vector.
Overall, this manuscript is a good-quality technical report/guideline, but it does not meet the bar for a publishable journal article.
Reviewer 2 Report
Comments and Suggestions for Authors
URGENT!!!!! References: The numbering system is completely nonsensical. Please order the references so that the reader can follow.
----------------------------------------------------------
Line 38: There is a rogue “i” in this line that should be removed.
Line 64: Reference number 5 lists the name of the vaccine but the authors use, “Merck’s Ebola vaccine.” For specificity, please indicate the name of the vaccine. Additionally, the italicized Ebola is meant to indicate the binomial nomenclature, or taxonomy, and is not necessary in this setting.
Line 65-70: Run on sentence; please revise for clarity.
Lin3 72: Reference number 6 is a link to the agency’s webpage and displays no immediate information regarding the countermeasures mentioned in the text of the manuscript. Please adjust the reference to be more specific for the vaccines mentioned.
Line 75-78: The italicized binomial nomenclature is unnecessary here as the vaccines are developed for a real-world virus and not the theoretical name of the species. For example, Ebola virus (EBOV) would be used here because when you refer to EBOV in the future you refer to the real virus and not the theoretical species.
Line 91: Authors use “Merck’s” again here, but this is a colloquialism and should be changed to reflect the full company name for specificity. Other places in the manuscript include names for companies and non-profits but it is unclear what or who they are.
Line 103: IAVI is no longer fully named International AIDS Vaccine Initiative, they just go by IAVI.
Line 106: The sentence beginning with “Other vectors…” should be the start of a new paragraph as the topic of discussion has changed from VSV to adenoviral vectors.
Line 150: Without the aid of the reference due to the numbers it is not clear if the authors mean by “single construct.” Please clarify this statement about the single construct.
Line 178-180: References for these specific cell lines are not included. Please include references for the reader.
Line 227: EUA is not spelled out as an acronym. Please spell out the first use of an acronym.
Line 261: References end at 45. Reference numbers need to be adjusted for the reader.
Line 275: CDMO is not spelled out as an acronym. Please spell out the first use of an acronym.
Line 297-305: Each of the platforms listed previously in the manuscript were discussed regarding maximum achieved status; for example, ERVEBO is licensed, and the Sabin adeno vector is at Phase 2 studies, etc. The authors introduce saRNA vaccines in the Discussion section and make no mention of where these vaccines stand across the globe in various stages of research, safety, and regulatory review. Please address the current research status of this platform compared to the other platforms.
General comment: The safety profile every vector listed throughout the manuscript should be discussed with respect to the risks associated with the indicated pathogen. Please make the distinction for use of novel technology vs. previously licensed technology and the scenarios where abandoning the previously licensed technology, as the authors mention is beneficia, should be abandoned.
Lines 317-338: There is no distinguishable meaning from this paragraph or the bullet points. If the reviewer understands correctly, you should know what you want to make, where to make it, and how much they can make. There is no mention of how the different technologies mentioned in the manuscript harmonize or dissonate with the manufacture process. The authors should address this lack of information by providing a substantive summary of EACH vector’s traits to consider for manufacture.
General comment: The authors make no comparison of vector use with respect to the target population. For example, if you were to use adenovirus 5, the general population is approximately 80% seropositive. The previous exposure to the virus might significantly affect the vector’s ability to deliver the desired antigen. The “reuseability” of these platforms would have direct impact on challenges and opportunities in manufacturing; for example, if the pathogen presents a high risk for mortality, perhaps choosing a vector for which a large portion of the population is seropositive, may be inappropriate as the “take” level would be insufficient. Additionally, if you were addressing two separate wide-spread, low mortality risk pathogens (think flu and covid), designing them based on the same vector might prove problematic as those that get the flu shot might not respond well when later they get the COVID shot, or vice versa. In this reviewer’s opinion, pre-existing immunity and manufacture strategy would go hand in hand. Please address how pre-existing immunity for each of these vectors may have a role in your manufacturing strategy and what roles pre-existing immunity might play into the regulatory review process, given the benefit of having a previously licensed technology but the detriment of each use precluding future use.
General comment: Without the appropriate reference labels, it’s hard to tell, but it’s the opinion of the reviewer that there is a lack of survey across previously licensed products. ERVEBO is provided by the authors as an example of licensed technology but there is no mention of other platforms despite the authors’ claim that prior licensure might provide a benefit. In fact, the reference to the HHS website for the different vectors in vaccinology is a bit sterile. That webpage provides no additional information nor discusses the reasons for which those vectors were used, for what they were used, or what their status might be across the product landscape (in and out of biodefense) as it speaks to the easier path toward licensure the authors mention as a decision point. Please include appropriate discussion/reference of the vectors and their status in the regulatory process.
General comment: There is no discussion of payload amongst the vectors. The vector delivers the payload, antigen to create an immune response; comparison of these differing carrying capacities would directly impact manufacturing challenges and opportunities. Some vectors might be heavier lifters than others (i.e. carrying more antigens), while some might be easier to manipulate or attached to less cost, both biologically and fiscally; the authors should mention of the capacity for each vector and how that capacity, or cost, might influence manufacture. Please include a discussion of the different vector payloads as these carrying capacities are essential for the reader to understand the challenge and opportunities in manufacturing.
General comment: Similar to payload, there is no discussion of each vector’s adaptability to various purification and storage techniques. Specific properties of each of the vectors might prove more or less beneficial for processes like lyophilization for storage; additionally, these properties might favor filtration over centrifugation, for example. Please discuss the qualities of the vectors chosen by BARDA in their manufacturing decisions.
General comment: BSL2 is very common and how these conditions are set is based on the BMBL. It would be appropriate for the authors to comment on the exact properties of the BSL2 laboratory that make manufacturing difficult in a piece regarding manufacturing challenges. Please discuss what about these containment conditions is difficult for manufacture and include appropriate reference(s).
Author Response
Reviewer 2
- Line 38: There is a rogue “i” in this line that should be removed.
Response: Thank you. The edit has been made.
- Line 64: Reference number 5 lists the name of the vaccine but the authors use, “Merck’s Ebola” For specificity, please indicate the name of the vaccine. Additionally, the italicized Ebola is meant to indicate the binomial nomenclature, or taxonomy, and is not necessary in this setting.
Response: Thank you. The edit has been made.
- Line 65-70: Run on sentence; please revise for clarity.
Response: Some additional punctuation was added to aid in clarity while aligning to internal requirements to note BARDA’s full affiliation and mission.
- Lin3 72: Reference number 6 is a link to the agency’s webpage and displays no immediate information regarding the countermeasures mentioned in the text of the manuscript. Please adjust the reference to be more specific for the vaccines mentioned.
Response: Thank you. An alternate citation that goes directly to the CBRN Vaccine program is now provided.
- Line 75-78: The italicized binomial nomenclature is unnecessary here as the vaccines are developed for a real-world virus and not the theoretical name of the species. For example, Ebola virus (EBOV) would be used here because when you refer to EBOV in the future you refer to the real virus and not the theoretical species.
Response: Thank you. The edit has been made.
- Line 91: Authors use “Merck’s” again here, but this is a colloquialism and should be changed to reflect the full company name for specificity. Other places in the manuscript include names for companies and non-profits but it is unclear what or who they are.
Response: Thank you. The document has been edited to not use company names.
- Line 103: IAVI is no longer fully named International AIDS Vaccine Initiative, they just go by IAVI.
Response: Thank you. The edit has been made.
- Line 106: The sentence beginning with “Other vectors…” should be the start of a new paragraph as the topic of discussion has changed from VSV to adenoviral vectors.
Response: Thank you. Edit made to make this the start of a new paragraph.
- Line 150: Without the aid of the reference due to the numbers it is not clear if the authors mean by “single construct.” Please clarify this statement about the single construct.
Response: Thank you. The document has been edited to use the term “single vector” rather than “single construct” to improve clarity.
- Line 178-180: References for these specific cell lines are not included. Please include references for the reader.
Response: Thank you. References have been added.
- Line 227: EUA is not spelled out as an acronym. Please spell out the first use of an acronym.
Response: Thank you. EUA is first spelled out in Section 2, paragraph 3
- Line 261: References end at 45. Reference numbers need to be adjusted for the reader.
Response: Thank you. The references have been adjusted.
- Line 275: CDMO is not spelled out as an acronym. Please spell out the first use of an acronym.
Response: Thank you. Edited to use CMO acronym.
- Line 297-305: Each of the platforms listed previously in the manuscript were discussed regarding maximum achieved status; for example, ERVEBO is licensed, and the Sabin adeno vector is at Phase 2 studies, etc. The authors introduce saRNA vaccines in the Discussion section and make no mention of where these vaccines stand across the globe in various stages of research, safety, and regulatory review. Please address the current research status of this platform compared to the other platforms.
Response: Thank you for your suggestion regarding additional detail on saRNA platform. While we thought it was important to acknowledge the potential of this technology – particularly in regards to the potential for single dose protection akin to viral vectors – it is not the primary focus of the manuscript. However, we have taken your suggestion into account and expanded on the development stage (preclinical) of the referenced filovirus study that demonstrated single dose protection as well as the development stage for COVID-19 and flu saRNA vaccines.
- General comment: The safety profile every vector listed throughout the manuscript should be discussed with respect to the risks associated with the indicated pathogen. Please make the distinction for use of novel technology vs. previously licensed technology and the scenarios where abandoning the previously licensed technology, as the authors mention is beneficia, should be abandoned.
Response: Thank you for the comments. Several references have been added that where BARDA has previously discussed safety and reactogenicity of viral vectors, illustrate the risk-benefit considerations, and product specific Target Product Profiles.
- Lines 317-338: There is no distinguishable meaning from this paragraph or the bullet points. If the reviewer understands correctly, you should know what you want to make, where to make it, and how much they can make. There is no mention of how the different technologies mentioned in the manuscript harmonize or dissonate with the manufacture process. The authors should address this lack of information by providing a substantive summary of EACH vector’s traits to consider for manufacture.
Response: The manuscript has been reframed in key parts to convey the intention of manuscript more clearly – which is to communicate the important role that viral vector vaccines play in BARDA’s CBRN portfolio focused on very high-consequence threats (e.g., filoviruses) and key considerations that inform a strategic approach to ensure this vaccine portfolio – and viral vector vaccine capacity – is sustainable. The author’s goal is to relay to stakeholders how BARDA is thinking about this important preparedness question. The noted section and bullets represent potential portfolio approaches to sustain viral vector manufacturing capacity for high-lethality infectious disease threats within the BARDA CBRN mission space. Edits are made to make it clear these are points BARDA is considering for the CBRN vaccine portfolio. Further, we have added additional technical details per the reviewer recommendations.
- General comment: The authors make no comparison of vector use with respect to the target population. For example, if you were to use adenovirus 5, the general population is approximately 80% seropositive. The previous exposure to the virus might significantly affect the vector’s ability to deliver the desired antigen. The “reuseability” of these platforms would have direct impact on challenges and opportunities in manufacturing; for example, if the pathogen presents a high risk for mortality, perhaps choosing a vector for which a large portion of the population is seropositive, may be inappropriate as the “take” level would be insufficient. Additionally, if you were addressing two separate wide-spread, low mortality risk pathogens (think flu and covid), designing them based on the same vector might prove problematic as those that get the flu shot might not respond well when later they get the COVID shot, or vice versa. In this reviewer’s opinion, pre-existing immunity and manufacture strategy would go hand in hand. Please address how pre-existing immunity for each of these vectors may have a role in your manufacturing strategy and what roles pre-existing immunity might play into the regulatory review process, given the benefit of having a previously licensed technology but the detriment of each use precluding future use.
Response: Thank you, the authors agree with this comment and address the point of preexisting vector immunity in the last paragraph of section 2; however, the point has now been reiterated in the discussion section for clarity.
- General comment: Without the appropriate reference labels, it’s hard to tell, but it’s the opinion of the reviewer that there is a lack of survey across previously licensed products. ERVEBO is provided by the authors as an example of licensed technology but there is no mention of other platforms despite the authors’ claim that prior licensure might provide a benefit. In fact, the reference to the HHS website for the different vectors in vaccinology is a bit sterile. That webpage provides no additional information nor discusses the reasons for which those vectors were used, for what they were used, or what their status might be across the product landscape (in and out of biodefense) as it speaks to the easier path toward licensure the authors mention as a decision point. Please include appropriate discussion/reference of the vectors and their status in the regulatory process.
Response: Thank you, your point is noted. A table has been added to more clearly note current BARDA viral vector vaccine programs. This paper has been reframed to discuss how viral vectors are used in a portfolio of vaccines against CBRN threats. BARDA funds vaccine development programs usually starting at a level of development at a Phase 1 or Phase 2 clinical study. Thus, BARDA does not typically play a role in the rationale for selection of vectors at the earliest discovery/preclinical stage of vaccine development, but rather considers advanced candidates with data indication potential to meet key program priorities.
- General comment: There is no discussion of payload amongst the vectors. The vector delivers the payload, antigen to create an immune response; comparison of these differing carrying capacities would directly impact manufacturing challenges and opportunities. Some vectors might be heavier lifters than others (i.e. carrying more antigens), while some might be easier to manipulate or attached to less cost, both biologically and fiscally; the authors should mention of the capacity for each vector and how that capacity, or cost, might influence manufacture. Please include a discussion of the different vector payloads as these carrying capacities are essential for the reader to understand the challenge and opportunities in manufacturing.
Response: Thank you, for the comments. The authors agree that this is a meaningful technical point. However, this level of deep technical detail is considered outside the scope of this manuscript that is focused on strategic considerations that inform an approach to sustaining the BARDA viral vector vaccine portfolio in the face of a very limited commercial market given the threat space.
- General comment: Similar to payload, there is no discussion of each vector’s adaptability to various purification and storage techniques. Specific properties of each of the vectors might prove more or less beneficial for processes like lyophilization for storage; additionally, these properties might favor filtration over centrifugation, for example. Please discuss the qualities of the vectors chosen by BARDA in their manufacturing decisions.
Response: Thank you. Please see response above. BARDA funds vaccine development programs usually starting at a level of development at a Phase 1 or Phase 2 clinical study. Thus, BARDA does not play a role in the rationale for selection of vectors for vaccine development for any given program. This paper has been reframed in key parts to more clearly convey the intention of manuscript – which is to communicate the important role that viral vector vaccines play in a portfolio focused on very high-consequence threats (e.g., filoviruses) and key considerations that inform a strategic approach to ensure this vaccine portfolio – and viral vector vaccine capacity – is sustainable. The author’s goal is to relay to stakeholders how BARDA is thinking about this important preparedness question. Further, we have added additional technical details per the reviewer recommendations.
- General comment: BSL2 is very common and how these conditions are set is based on the BMBL. It would be appropriate for the authors to comment on the exact properties of the BSL2 laboratory that make manufacturing difficult in a piece regarding manufacturing challenges. Please discuss what about these containment conditions is difficult for manufacture and include appropriate reference(s).
Response: Thank you for this note. Some edits have been made for clarity. Further, in section 3 the containment challenges live virus manufacturing presents have been noted. Additionally, while the authors agree that BSL2 laboratory space is common, manufacturing facilities (CDMOs) with BSL2/live virus manufacturing capacity – whether for bulk drug substance or fill/finish to final drug product – is very limited. Further viral vector vaccines for BARDA priority threats that have no commercial market compete for space with other therapeutic areas that have, which ties to the sustainability of the BARDA portfolio.
Reviewer 3 Report
Comments and Suggestions for Authors
In this review, Parish et al. describe various vaccine technologies, highlight the need to improve manufacturing capabilities domestically, and showcase some of the contributions the Biomedical Advanced Research and Development Authority (BARDA) has made in this space. For clarity and ease of reading, consider addressing the following comments to improve the overall text.
- Line 38: remove the “i" between “such as” and “immunogenicity”.
- Line 289: Remove “].”
- There are two Section 3’s and two Section V’s.
- Numbers in references are disorganized. Once fixed, confirm in-text bracketed reference numbers are correct.
- BARDA’s portfolio of vector vaccines: Section header indicates the text to follow will discuss BARDA’s direct involvement and current portfolio of vectored vaccines. However, the text that follows discusses several commercially/privately licensed or developed vaccine candidates. Clarify that the vaccines discussed (ERVEBO, VSV SUDV, Zabdeno and Mvabea, ChAd3-MARV and ChAd3-SUDV, etc.) in the first two paragraphs were directly supported by BARDA.
- Consider combining the second and third paragraphs from the introduction with the first and second paragraph from “BARDA’s portfolio of vector vaccines” to a new section 2 that describes Viral Vector Vaccines. A new header would be required for Project NextGen.
- The authors briefly mention in their conclusion potential challenges surrounding licensing agreements. Consider including more details on how, as a US-government funded research organization part of HHS, the benefit of patenting/licensing technology developed through a BARDA pipeline extends globally. Government patents can often be licensed more broadly and are less likely to be driven by profit motives.
- The use of self-amplifying RNA vaccines is briefly discussed in relation to high consequence pathogens and the authors suggest further investigation. Numerous saRNAs have been developed and shown to be quite successful in animal studies (100% protection against lethal EBOV challenge with single dose saRNA (doi 10.1016/j.ymthe.2022.10.011)). This and other studies could be used as a segue into BARDA’s current involvement in the field.
- Related to saRNA vaccines, I believe BARDA is actively working on intranasal saRNA vaccines against influenza. If true, this involvement could be described when mentioning the challenges of mRNA vaccine delivery through the mucosal barrier.
Author Response
Reviewer 3
In this review, Parish et al. describe various vaccine technologies, highlight the need to improve manufacturing capabilities domestically, and showcase some of the contributions the Biomedical Advanced Research and Development Authority (BARDA) has made in this space. For clarity and ease of reading, consider addressing the following comments to improve the overall text.
- Line 38: remove the “i" between “such as” and “immunogenicity”.
Response: We thank the reviewer for their comments. This edit has been made.
- Line 289: Remove “].”
Response: Thank you. Removed.
- There are two Section 3’s and two Section V’s.
Response: Thank you. The section numbering has been fixed.
- Numbers in references are disorganized. Once fixed, confirm in-text bracketed reference numbers are correct.
Response: Thank you. The references have been redone.
- BARDA’s portfolio of vector vaccines: Section header indicates the text to follow will discuss BARDA’s direct involvement and current portfolio of vectored vaccines. However, the text that follows discusses several commercially/privately licensed or developed vaccine candidates. Clarify that the vaccines discussed (ERVEBO, VSV SUDV, Zabdeno and Mvabea, ChAd3-MARV and ChAd3-SUDV, etc.) in the first two paragraphs were directly supported by BARDA.
Response: Thank you. Your point is noted. A table has been added to more clearly note current BARDA viral vector vaccine programs.
- Consider combining the second and third paragraphs from the introduction with the first and second paragraph from “BARDA’s portfolio of vector vaccines” to a new section 2 that describes Viral Vector Vaccines. A new header would be required for Project NextGen.
Response: Thank you for your comments. A new header for Project NextGen has been added and introduction reorganized.
- The authors briefly mention in their conclusion potential challenges surrounding licensing agreements. Consider including more details on how, as a US-government funded research organization part of HHS, the benefit of patenting/licensing technology developed through a BARDA pipeline extends globally. Government patents can often be licensed more broadly and are less likely to be driven by profit motives.
Response: Thank you for this comment. Some additional context has been added to the reference sentence to clarify that the statement is referring to licensing agreements between smaller developers and larger corporate entities that own the IP. Since BARDA typically supports advanced development, developers are coming in with established IP or licensing agreements and BARDA does not own said patent rights.
- The use of self-amplifying RNA vaccines is briefly discussed in relation to high consequence pathogens and the authors suggest further investigation. Numerous saRNAs have been developed and shown to be quite successful in animal studies (100% protection against lethal EBOV challenge with single dose saRNA (doi 10.1016/j.ymthe.2022.10.011)). This and other studies could be used as a segue into BARDA’s current involvement in the field.
Response: Thank you. The recommended reference was added to the sentence that noted early preclinical studies that have demonstrated single dose protection for filoviruses with saRNA. As this manuscript is focused on viral vector vaccines, detailed discussion of BARDA’s saRNA is outside the technical scope of this manuscript.
- Related to saRNA vaccines, I believe BARDA is actively working on intranasal saRNA vaccines against influenza. If true, this involvement could be described when mentioning the challenges of mRNA vaccine delivery through the mucosal barrier.
Response: Thank you for this note. However, this manuscript is focused on discussing key considerations for a BARDA portfolio approach to ensuring sustainability of viral vector vaccines for priority threats. While this is an important technical consideration for mucosal RNA vaccines, this is outside the scope of this manuscript that has been reframed to communicate the important role that viral vector vaccines play in a portfolio focused on very high-consequence threats.
Round 2
Reviewer 1 Report
Comments and Suggestions for Authors
One minor comment: for the Table 1 in the revised manuscipt, please seperate RNA/DNA from (non)enveloped into two column. Whether enveloped or non-enveloped is belong to viral capsid, not belongs to under Genome.
Author Response
We thank the reviewer for their comments. A separate column to Table 1 has been added to indicate whether a virus is enveloped or non-enveloped.
Reviewer 2 Report
Comments and Suggestions for Authors
Second review general comment: The authors have responded to most all reviewer’s comments excellently! Thank you very much for addressing these concerns. There are only a couple, very small comments for consideration.
Reference 1: The reference contains no URL; it is not common to refer to a website and not add the address. Please add the address for the ease of the reader to navigate accurately to the referenced site.
Reference 7: The reference contains no URL; it is not common to refer to a website and not add the address. Please add the address for the ease of the reader to navigate accurately to the referenced site.
Reference 17: The reference is first used on line 101 referring to Ervebo, the VSV based vaccine for Ebola, in that sentence; however, the reference is about adenovirus vectors with no mention of Ebola in the text of the reference. Please accurately place this reference or delete it.
Line 131: The reference for this sentence is listed as “[308].” Please revise.
Reference 34: The reference contains no URL; it is not common to refer to a website and not add the address. Please add the address for the ease of the reader to navigate accurately to the referenced site.
Reference 35: The reference contains no URL; it is not common to refer to a website and not add the address. Please add the address for the ease of the reader to navigate accurately to the referenced site.
Lines 181-219: The authors added a substantial amount of text, possibly in response the previous review; however this text is not specifically attributable to any comment. The text is also ambiguous. For example, the authors state, “The high variability between products, titering methods, and viral genome load versus capsid exposure can be mitigated by use of reference standards that can reduce variability giving more precise and reliable quantification so that we can be more confident about the limit of detection.” No additional information is given about the variability between products, tittering methods, viral genome load versus capsid exposure, mitigation by use of reference standards, or what those references standards might be to provide more precise and reliable quantification. There is also no information given about the confidence of the limit of detection, methods used, or what those limits might be across the landscape. These broad terms without reference or discussion will leave the reader without answers. Please address these topics in the text to be more specific toward each subject introduced toward the manuscript’s goal of “Strategic and Technical Considerations in Manufacturing Viral Vector Vaccines for the Biomedical Advanced Research and Development Authority Threats.”
Lines 215-218: The authors state, “Moreover, opportunities for integrating analogous scaled down models across viral vectored vaccines to optimize process development enabling predication of process parameters and product quality can derisk the process and help identify cost efficient solutions.” This sentence seems out of place. this sentence runs the risk of meaning nothing to the reader given the lack of references, and missing specificity for the words:
-analogous scaled down models
-across viral vectored vaccines
-optimize process development
-enabling predication
-process parameters
-product quality
-derisk the process
-identify cost efficient solutions
The listed terms and vagueness should be specified, elaborated upon, defined, referenced, etc. If the authors are unwilling to do the previously mentioned then the sentence should be deleted from the text, in this reviewer’s opinion, as to not introduce confusion for reader. Please revise the text to specifically advise the reader toward the aim of the manuscript.
Reference 54: The reference contains no URL; it is not common to refer to a website and not add the address. Please add the address for the ease of the reader to navigate accurately to the referenced site.
Reference 55: The reference contains no URL; it is not common to refer to a website and not add the address. Please add the address for the ease of the reader to navigate accurately to the referenced site.
Comment 17 from first review: The authors have added half a sentence to the discussion, which is appreciated; however, there is no mention of preexisting immunity in the last paragraph of the second section, perhaps it was dropped in the new version. While improved, there is no specific language as to the impact of preexisting immunity on, “Strategic and Technical Considerations in Manufacturing…” Does basing multiple vaccines on the same platform cause a problem of preexisting immunity to the vector and does that preexisting immunity factor into new vectors requiring manufacture? If that’s the case does that manufacturing space from the previous vectors have any overlap with new vectors? The implication is contained within the authors’ words, “…as well as vector-specific technical considerations like potential for single dose protection and impact of preexisting vector immunity;” however, the exact fallout of those technical considerations is not explicitly stated. Please revert to the previous version of section 2, last paragraph, if there was dropped language, and please include a bullet point, specific words, or edit on the impact of preexisting immunity on, “Strategic and Technical Considerations in Manufacturing….”
Author Response
We thank the reviewer for their comments. Below is each comment with our response in blue.
Reference 1: The reference contains no URL; it is not common to refer to a website and not add the address. Please add the address for the ease of the reader to navigate accurately to the referenced site. We thank the reviewer for their comments. We have fixed the URL to no longer be an embedded link.
Reference 7: The reference contains no URL; it is not common to refer to a website and not add the address. Please add the address for the ease of the reader to navigate accurately to the referenced site. We thank the reviewer for their comments. We have fixed the URL to no longer be an embedded link.
Reference 17: The reference is first used on line 101 referring to Ervebo, the VSV based vaccine for Ebola, in that sentence; however, the reference is about adenovirus vectors with no mention of Ebola in the text of the reference. Please accurately place this reference or delete it. We thank the reviewer for their comments. We have corrected this error and fixed reference #17.
Line 131: The reference for this sentence is listed as “[308].” Please revise. We thank the reviewer for their comments. We have corrected this typo.
Reference 34: The reference contains no URL; it is not common to refer to a website and not add the address. Please add the address for the ease of the reader to navigate accurately to the referenced site. We thank the reviewer for their comments. We have fixed the URL to no longer be an embedded link.
Reference 35: The reference contains no URL; it is not common to refer to a website and not add the address. Please add the address for the ease of the reader to navigate accurately to the referenced site. We thank the reviewer for their comments. We have fixed the URL to no longer be an embedded link.
Lines 181-219: The authors added a substantial amount of text, possibly in response the previous review; however this text is not specifically attributable to any comment. The text is also ambiguous. For example, the authors state, “The high variability between products, titering methods, and viral genome load versus capsid exposure can be mitigated by use of reference standards that can reduce variability giving more precise and reliable quantification so that we can be more confident about the limit of detection.” No additional information is given about the variability between products, tittering methods, viral genome load versus capsid exposure, mitigation by use of reference standards, or what those references standards might be to provide more precise and reliable quantification. There is also no information given about the confidence of the limit of detection, methods used, or what those limits might be across the landscape. These broad terms without reference or discussion will leave the reader without answers. Please address these topics in the text to be more specific toward each subject introduced toward the manuscript’s goal of “Strategic and Technical Considerations in Manufacturing Viral Vector Vaccines for the Biomedical Advanced Research and Development Authority Threats.” We thank the reviewer for their comments. We agree that the insertion without the original context of the reviewer comment was somewhat confusion. We have inserted text to provide some context that the lack of standardization results in risks with production. And if assays and processes are standardized for a given insert, that may reduce risks for the platform as a whole.
  Lines 215-218: The authors state, “Moreover, opportunities for integrating analogous scaled down models across viral vectored vaccines to optimize process development enabling predication of process parameters and product quality can derisk the process and help identify cost efficient solutions.” This sentence seems out of place. this sentence runs the risk of meaning nothing to the reader given the lack of references, and missing specificity for the words:
               -analogous scaled down models
               -across viral vectored vaccines
               -optimize process development
               -enabling predication
               -process parameters
               -product quality
               -derisk the process
               -identify cost efficient solutions
The listed terms and vagueness should be specified, elaborated upon, defined, referenced, etc. If the authors are unwilling to do the previously mentioned then the sentence should be deleted from the text, in this reviewer’s opinion, as to not introduce confusion for reader. Please revise the text to specifically advise the reader toward the aim of the manuscript. We thank the reviewer for their comments. We have deleted this sentence.
Reference 54: The reference contains no URL; it is not common to refer to a website and not add the address. Please add the address for the ease of the reader to navigate accurately to the referenced site. We thank the reviewer for their comments. We have fixed this error.
 
Reference 55: The reference contains no URL; it is not common to refer to a website and not add the address. Please add the address for the ease of the reader to navigate accurately to the referenced site. We thank the reviewer for their comments. We have fixed the URL to no longer be an embedded link.
 
Comment 17 from first review: The authors have added half a sentence to the discussion, which is appreciated; however, there is no mention of preexisting immunity in the last paragraph of the second section, perhaps it was dropped in the new version. While improved, there is no specific language as to the impact of preexisting immunity on, “Strategic and Technical Considerations in Manufacturing…” Does basing multiple vaccines on the same platform cause a problem of preexisting immunity to the vector and does that preexisting immunity factor into new vectors requiring manufacture?  If that’s the case does that manufacturing space from the previous vectors have any overlap with new vectors? The implication is contained within the authors’ words, “…as well as vector-specific technical considerations like potential for single dose protection and impact of preexisting vector immunity;” however, the exact fallout of those technical considerations is not explicitly stated. Please revert to the previous version of section 2, last paragraph, if there was dropped language, and please include a bullet point, specific words, or edit on the impact of preexisting immunity on, “Strategic and Technical Considerations in Manufacturing….”
We thank the reviewer for their comments. A bullet in the discussion has been edited to address the considerations for pre-existing immune. The original discussion on pre-existing immunity was moved out of the discussion and into the body of the manuscript at the request of another reviewer.